# New Drugs for Human African Trypanosomiasis: A Twenty First Century Success Story

**DOI:** 10.3390/tropicalmed5010029

**Published:** 2020-02-19

**Authors:** Emily A. Dickie, Federica Giordani, Matthew K. Gould, Pascal Mäser, Christian Burri, Jeremy C. Mottram, Srinivasa P. S. Rao, Michael P. Barrett

**Affiliations:** 1Wellcome Centre for Integrative Parasitology, Institute of Infection, Immunity and Inflammation, University of Glasgow, Glasgow G12 8TA, UK; emily.dickie@glasgow.ac.uk (E.A.D.); federica.giordani@glasgow.ac.uk (F.G.); matthew.gould@glasgow.ac.uk (M.K.G.); 2Swiss Tropical and Public Health Institute, Socinstrasse 57, 4002 Basel, Switzerland; pascal.maeser@swisstph.ch (P.M.); christian.burri@swisstph.ch (C.B.); 3University of Basel, Petersplatz 1, 4000 Basel, Switzerland; 4York Biomedical Research Institute, Department of Biology, University of York, Wentworth Way, Heslington, York YO10 5DD, UK; jeremy.mottram@york.ac.uk; 5Novartis Institute for Tropical Diseases, 5300 Chiron Way, Emeryville, CA 94608, USA; srinivasa.rao@novartis.com

**Keywords:** human African trypanosomiasis, sleeping sickness, elimination, chemotherapy, fexinidazole, pafuramidine, acoziborole

## Abstract

The twentieth century ended with human African trypanosomiasis (HAT) epidemics raging across many parts of Africa. Resistance to existing drugs was emerging, and many programs aiming to contain the disease had ground to a halt, given previous success against HAT and the competing priorities associated with other medical crises ravaging the continent. A series of dedicated interventions and the introduction of innovative routes to develop drugs, involving Product Development Partnerships, has led to a dramatic turnaround in the fight against HAT caused by *Trypanosoma brucei gambiense*. The World Health Organization have been able to optimize the use of existing tools to monitor and intervene in the disease. A promising new oral medication for stage 1 HAT, pafuramidine maleate, ultimately failed due to unforeseen toxicity issues. However, the clinical trials for this compound demonstrated the possibility of conducting such trials in the resource-poor settings of rural Africa. The Drugs for Neglected Disease initiative (DNDi), founded in 2003, has developed the first all oral therapy for both stage 1 and stage 2 HAT in fexinidazole. DNDi has also brought forward another oral therapy, acoziborole, potentially capable of curing both stage 1 and stage 2 disease in a single dosing. In this review article, we describe the remarkable successes in combating HAT through the twenty first century, bringing the prospect of the elimination of this disease into sight.

## 1. Introduction

The current drugs used for human African trypanosomiasis (HAT) (Figure 1) have served their purpose for many years. The incidence of HAT is now at a historic low (fewer than 1000 cases reported in 2018 [1]). Two forms of the disease occur. The one found in West and Central Africa is caused by *Trypanosoma brucei gambiense*, and the other, found in East and Southern Africa, is caused by *Trypanosoma brucei rhodesiense*. The former causes a chronic disease, taking years between infection and death, while the latter may kill within weeks to months. Parasites injected into the bloodstream cause a stage 1 infection, where replication is primarily associated with blood and lymph. However, the parasites then invade other organs, including the central nervous system (CNS). Once replicating in the CNS, the disease progresses to stage 2, where many of the symptoms of sleeping sickness become manifest. The current drugs suffer many drawbacks [2]. For stage 1 disease, either suramin or pentamidine is used for HAT caused by *T. b. rhodesiense* and *T. b. gambiense*, respectively. Both drugs must be given by injection for a prolonged period, and both carry a risk of adverse events. For stage 2 disease, the highly toxic melarsoprol is still the treatment of choice for rhodesiense HAT. Melarsoprol causes an encephalopathic syndrome that is fatal in up to one in twenty people taking the drug [3]. For gambiense HAT, the past decade has seen the introduction of a combination therapy. Intravenous eflornithine is given for 10 days alongside oral nifurtimox for 14 days [4]. 

The combination is relatively safe and efficacious. However, the delivery of kilogram quantities of eflornithine and many liters of sterile saline brings substantial logistical difficulty. For melarsoprol, resistance was a growing problem in the early 2000s [5], and for eflornithine [6] and nifurtimox [7] independently, resistance can be readily selected in the laboratory. There is no doubt that better drugs for use against HAT are required [8]. 

The first two decades of the twenty first century can be seen as a major success story in regards to intervention against this neglected tropical disease. HAT was running out of control in the late twentieth century, with an estimated 300,000 people infected [9]. In response, the international community launched several key initiatives, which can be seen as having converged to turn the tide. Consequently, the twenty first century has witnessed a dramatic change in the trajectory of HAT.

In 1999, Médecins Sans Frontières (MSF) initiated their “Access” campaign to encourage a re-engagement of the pharmaceutical industry with neglected diseases, including HAT [10]. A key study [11] revisited the administration regimen of melarsoprol based on pharmacokinetic data. An effective 10-day administration protocol, which diminished hospitalization time for patients [12], albeit without increasing safety, was introduced to great effect [13]. This IMPAMEL program (“improved application of melarsoprol”) was crucial in implementing the first modern clinical trials on HAT, and also in demonstrating the feasibility of conducting trials in extremely resource-limited conditions according to Good Clinical Practice, laying the foundation for future developments. 

Eflornithine had been shown to be far safer than melarsoprol as treatment for stage 2 disease [14]. Yet, by the late 1990s, no pharmaceutical company was prepared to make this compound for HAT treatment. However, when it was discovered that the same compound could prevent the growth of unwanted facial hair in women, a number of drug companies saw an opportunity to market the compound for this purpose [15]. MSF, already campaigning for access to essential medicines, were able to make a compelling case that society needed to rethink drug discovery paradigms for neglected diseases [16]. Aventis (now Sanofi) were persuaded to develop the drug and donate it at no cost to the World Health Organization (WHO) for distribution in Africa. Millions of dollars were also provided by Aventis/Sanofi to WHO, who could now develop new screening and intervention programs. The Bill and Melinda Gates Foundation selected HAT to be one of the first diseases they targeted through the Consortium of Parasitic Drug Development (CPDD) [17], and the Drugs for Neglected Diseases initiative (DNDi) was founded through MSF [18] to seek new drugs for diseases including HAT. In diagnostics, the Foundation for Innovative New Diagnostics (FIND) sought novel ways of improving our ability to detect HAT patients [19], and new ways of combating the tsetse fly were rolled out too [20]. A number of pharmaceutical companies also regained an interest in HAT, including GlaxoSmithKline (GSK) through their Tres Cantos Open Lab foundation [21] and the Novartis Institute for Tropical Diseases (NITD) [22]. Small companies too, such as Immtech Pharmaceuticals Inc., Scynexis and Anacor Pharmaceuticals Inc. in the USA, also raised investment to develop new drugs against HAT. The first twenty years of the twenty first century have now seen the clinical trials and ultimate failure of a new orally available diamidine prodrug [23], the registration of the first all-orally available therapy against stage 2 disease [24], and the entry into clinical trials of a compound that may cure stage 2 HAT with a single oral administration [25]. This article outlines the successes seen in the development of new drugs for HAT in the twenty first century.

## 2. Pafuramidine—A New Paradigm in Anti-Trypanosomal Drug Development

Among the currently used drugs for HAT is pentamidine, a diamidine that was introduced in the 1940s, and has been the mainstay in the treatment of stage 1 gambiense HAT for nearly 80 years [26]. Another diamidine, diminazene, is used in treating veterinary trypanosomiasis [27]. The diamidines are di-cations, with positive charges at either end, which renders them highly polar, precluding bioavailability if taken orally [28]. Pentamidine is typically given by intramuscular injection for seven days. Das and Boykin showed in the 1970s that methoxy-derivatives of other diamidines acted as orally available prodrugs, and the methoxy group metabolized back to the amidine systemically [29]. 

The Bill and Melinda Gates Foundation was founded in 2000, a time when the resurgence of HAT was at its pinnacle. Among the Foundation’s earliest supported projects was the development of the diamidine methoxy-prodrug approach towards new drugs for HAT through the CPDD.

DB289 (Figure 2) (pafuramidine maleate; 2,5-bis(4-amidinophenyl)-furan-bis-*O*-methylamidoxime), the methoxy product of furamidine (2,5-bis(4-amidinophenyl)-furan), in which the benzamidine moieties are separated by a furan ring, emerged as the lead compound for progression.

Furamidine is highly polar and unable to traverse lipid bilayers without the assistance of transporters (see later). Pafuramidine, however, has much greater capacity to diffuse across membranes [30], including the intestinal epithelium, giving it considerable oral bioavailability. Once systemic, it is metabolized via various cytochrome P450 enzymes and cytochrome b5 reductase [31,32,33,34,35,36] to furamidine. Preclinical safety and efficacy results [37,38] were sufficient to enable the passage to clinical trials.

### 2.1. Clinical Trials of Pafuramidine

Ultimately pafuramidine failed in clinical trials [23,39]. However, those trials were of great importance, not only in testing the efficacy of this promising lead compound, but also in informing on any criteria required to conduct clinical trials for regulatory purposes in a cohort of patients in difficult places to work. These trials also helped to establish protocols for patient seeking, screening, drug administration, record keeping and patient follow-up post-treatment. The very fact that over 350,000 individuals were screened in the quest for patients to include in the trials had a major impact on the incidence of the disease. Patients testing positive but failing inclusion criteria (i.e., not clearly in stage 1 disease), were treated with other drugs appropriate to their stage. 

An original phase I safety study (unpublished), carried out in healthy Caucasian volunteers in Germany in 2001 showed the drug to be well tolerated in single and multi-dose testing (100 mg twice per day ( b.i.d.)) and up to 600 mg in a single dose. 

A phase IIa study [39] in Viana, Angola and the CDTC Maluku, Democratic Republic of the Congo (DRC) with 32 patients showed that 100 mg of pafuramidine orally twice a day for 5 days yielded loss of all visible trypanosomes in blood 24 h after treatment cessation in 93% of cases, compared to 100% efficacy with 7-day pentamidine treatment. However, prolonged follow up began to pick up enhanced relapse rates from this five-day dosing (by 24 months, only 67% of pafuramidine-treated patients were considered cured). 81 patients had entered a phase IIb trial, at the same 5-day dosing (40 for pafuramidine and 41 for pentamidine) [39] by this time. However, the high relapse rate after prolonged follow up in the phase IIa-1 trial prompted a decision to use a 10-day dosing of 100 mg pafuramidine b.i.d. instead, and 30 more patients were enrolled for this phase IIa-2 trial. The dose was well tolerated over 10 days, and was less toxic than pentamidine [39]. The 10-day dosing of pafuramidine gave 93% cure at 3 months follow up, and was selected for a Phase III trial. 

The Phase III trial [23], conducted in several centers in the Democratic Republic of the Congo, Angola, and South Sudan recorded a cure rate of 89% at 12 months follow up. The major safety concern noted in these treatment trials was a 7% incidence of increased liver enzymes; however, the incidence of this adverse event was substantially less than in the comparator group treated with pentamidine (77%). A significantly lower percentage (2% vs. 9%) of patients treated with pafuramidine experienced treatment emergent adverse events detected by renal and urinary tract investigations and urinalyses compared to those under pentamidine. Overall, three subjects in the pafuramidine treatment groups were reported to have serious adverse events that could be considered acute renal failure or insufficiency classified as possibly associated with study drugs [23]. 

As the phase II/III trial was progressing, a further phase I safety trial was conducted in South Africa, necessitated by the increase in the duration of drug administration and the limited number of patients exposed to pafuramidine (unpublished). One hundred healthy sub-Saharan African adult volunteers were treated with 100 mg pafuramidine b.i.d. for 14 days. Generally, a mild and reversible increase of liver enzymes was observed in multiple subjects, which had been anticipated [23]. However, 6 of 100 subjects then experienced delayed renal insufficiency with unclear etiology. 

The delayed action might indicate a possible immunological role; HAT is itself immunosuppressive, which could explain why the incidence of kidney injury in treated patients was much lower. Another hypothesis was a genetic origin across a diverse group of individuals, which was supported by a retrospective study on a panel of 34 genetically distinct mouse strains [40]. Urinary secretion of the kidney injury molecule-1 (KIM-1) protein was used as a marker of proximal tubule injury in the kidney. Only a subset of the mice revealed elevated KIM-1. Genetic association studies then showed several genes with alleles associated with the KIM-1 levels. This included PCSK5, a serine peptidase associated with lipid and cholesterol metabolism, and another cholesterol-associated enzyme, sterol O-acyltransferase, among others. Definitive explanations for the kidney injury mechanism, however, remain obscure. DNA from the subjects involved in the clinical trial was not available to determine whether the same genes were associated to kidney damage in humans. 

The appearance of unanticipated toxicity with pafuramidine led to the end of clinical development and also a cessation of other activities around the diamidine project. This was unfortunate, since the project had also identified several aza-analogs, including DB820 and DB829 (alongside their respective methoxyamidine prodrugs DB844 and DB868,), which were effective in mouse [41] and vervet monkey (*Chlorocebus pygerythrus*) models of stage 2 infection [42]. Significantly, DB829 accumulated far less (>10 fold) inside mammalian cells than furamidine [43], pointing to a potentially improved safety profile. For HAT, however, the successes that have occurred with other compounds (see below) indicate that the diamidines are unlikely to be resurrected for future work. However, because of their diverse and intriguing biological activities [44], it is likely to see them re-emerging for other conditions in the future [45].

### 2.2. Pafuramidine: Mode of Action and Resistance Risk

Furamidine, like other diamidines, binds to the minor groove of the DNA double helix [28,45]. The ability to bind is structure-dependent, and intriguing work has aimed to tailor this binding to specific DNA sequences, including regulatory elements [44], with a view to controlling gene expression in, for example, human cancer cells. Furamidine is fluorescent, and its accumulation in trypanosomes shows binding to both the kinetoplast (mitochondrial DNA) and nuclear DNA, as well as its accumulation in vesicles assumed to be acidocalcisomes [46]. The compound accumulates to millimolar concentration, even in cells exposed to low micromolar concentrations over 24 hours [46]. Even a brief exposure (<5 min) to 32 micromolar of the compound, followed by wash out, provoked death 48 hours later [47]. At 3.2 micromolar, exposure time rose to 1 hour to achieve this slow commitment to death. Lanteri also showed a profound decrease in the mitochondrial membrane potential associated with the addition of furamidine to *T. brucei* [48]. 

The F1Fo ATP synthase complex is an essential mitochondrial complex in trypanosomes. In procyclic forms it acts in a classical fashion, producing ATP as part of the electron transport chain. In bloodstream forms, however, it acts in reverse: consuming ATP in order to maintain the essential mitochondrial membrane potential. All of the F1Fo ATPase subunits are encoded in the nucleus, apart from subunit A6, which is encoded in the kinetoplast. Mutations to the nuclearly-encoded gamma subunit (e.g., L262P substitution or an alanine deletion at amino acid position 281) allow the mitochondrial membrane potential to be generated in bloodstream form trypanosomes without subunit A6, rendering the kinetoplast redundant in this life cycle stage. The kinetoplast can be removed from these cells without apparent impact on viability [49]. The ATPase gamma mutants (now kinetoplast independent) are hundreds of fold resistant to phenanthridine compounds; intriguingly however, only very minor resistance (~ 3-fold) was shown for furamidine [50]. This would indicate that, in contrast to phenanthridines, it is not the kinetoplast per se that is the target of furamidine.

Diamidines also act against yeast mitochondria [51] with *Saccharomyces cerevisiae* that are fermenting glucose, rather than respiring using mitochondrial substrates (e.g., glycerol), being 200-fold less sensitive to pentamidine [52].

Mammalian cells too are vulnerable to diamidines. However, the hypersensitivity of trypanosomes to this class is due to their ability to accumulate diamidines to very high concentrations across their plasma membrane through specific transporters. The P2 aminopurine transporter, *Tb*AT1, which normally carries adenosine and adenine, was the first transporter characterized to carry pentamidine [53]. Subsequently, a high affinity pentamidine transporter [54], later revealed as the aquaglyceroporin *Tb*AQP2 [55], was shown to play a dominant role in uptake [56]. A low affinity pentamidine transporter, whose physiological role remains elusive, was also shown to transport pentamidine [54]. Loss of *Tb*AT1 yields only low-level resistance to pentamidine, while loss of *Tb*AQP2 yields high level resistance [56]. Pentamidine, however, is an exception among the trypanocidal diamidines, having a highly flexible central linker chain. The shorter derivatives (e.g., diminazene, and furamidine plus its aza-analogs), primarily use the *Tb*AT1 aminopurine transporter for uptake [57], and its loss gives high level resistance. Discovering a specific motif found on diamidines, aminopurines and also melaminophenyl arsenicals, all of which can enter via the *Tb*AT1 P2 transporter [58], led to several efforts to create new selective trypanocides to enter via the same carrier protein [59,60,61]. 

Diamidines can enter mammalian cells and also cross the blood–brain barrier (BBB) [62]. The human facilitative organic cation transporter 1 (OCT1) is one transporter capable of transporting pentamidine and other diamidines in mammalian cells [63,64]. ATP-dependent pumps (e.g., the P-glycoprotein, P-gp) appear to also be able to efflux pentamidine [62], but not furamidine, nor DB829 [65]. 

## 3. Fexinidazole: the First Oral Treatment for HAT

In November 2018, The European Medicines Agency’s Committee for Medicinal Products for Human Use (CHMP) offered a positive opinion on the use of oral fexinidazole for the treatment of both stage 1 stage 1 and 2 gambiense HAT [66]. In December 2018 the DRC, the epicenter of the disease, issued marketing authorization allowing use of the drug. Fexinidazole is given once per day for ten days, involving a four-day loading dose of 1.8 g per day followed by six days at 1.2 g per day with 600 mg tablets. The clinical development of fexinidazole is described in an accompanying article in this volume [67]. 

### 3.1. Pre-Clinical Development of Fexinidazole

The pathway for fexinidazole to the clinic was both long and disrupted. The drug, a 2-substituted 5-nitroimidazole, was originally synthesized as part of a program seeking anti-infectives by Hoechst in the 1970s [68]. Frank Jennings and George Urquhart at the University of Glasgow extended trypanocidal testing in the 1980s [69]. They could not demonstrate a full cure against the stage 2 disease model in mice which they had developed using the *T. brucei brucei* GVR35 strain which develops a slowly progressing disease in which the parasite enters the brain and establishes a CNS infection, prior to having killed the mice in an acute infection, as most laboratory strains of *T. brucei brucei* do. Administering fexinidazole as a monotherapy (given once a day for four days at 250 mg/kg), it cured 11/14 mice [69], although sequential treatment with a single dose of suramin (20 mg/kg) or diminazene (40 mg/kg), followed by four consecutive daily doses of fexinidazole, was curative. In the case of the suramin followed by a fexinidazole regime, a single dose of suramin, followed by four daily doses of just 30 mg/kg fexinidazole, was curative [69]. It was later shown [70] that five days of fexinidazole monotherapy at 200 mg/kg was curative (7/8 mice) in the stage 2 model, showing the necessity for prolonged exposure.

The finding that many nitroheterocycles were genotoxic and potentially carcinogenic led to a general aversion to this class of molecule for its use as pharmaceuticals for several decades. However, nifurtimox, another nitroheterocycle, entered the anti-trypanosomal armamentarium as a combination partner with eflornithine [4]. The nitroheterocycle known as megazol also showed interesting trypanocidal activity [71], whilst other nitroheterocycles such as PA-824 (now registered as pretomanid for tuberculosis [72]) came to the fore. 

Thus, interest in trypanocidal nitroheterocycles was rekindled. A series of 830 nitroimidazoles and related compounds were tested at the Swiss Tropical and Public Health Institute against *T. brucei*. *Trypanosoma cruzi*, *Leishmania donovani*, and mammalian cells *in vitro*. The most active and selective molecules were evaluated in secondary *in vitro* assays and in the mouse models of acute or chronic trypanosomiasis. Based on these data, fexinidazole was singled out as the most promising candidate for further development against HAT [70,73], and a dossier of information was compiled to support the safety and efficacy of the drug [67,74]. 

### 3.2. Fexinidazole: Mode of Action and Resistance Risk

Fexinidazole is a prodrug. Its activity depends upon two consecutive electron reductions of the NO_2_ group by an NADH-specific nitroreductase (*Tb*NTR1) [75]. Diminished NTR activity leads to resistance to fexinidazole [76] and cross-resistance to other nitroheterocycles, including nifurtimox. The fate of the nitro-reduced product is unknown. An orthologous nitroreductase enzyme is also found in *T. cruzi* [75], where it is responsible for the activation of clinically used nitroheterocycles (e.g., benznidazole and nifurtimox). In the case of nifurtimox, *T. cruzi* was shown to reduce the compound to a highly reactive species [77]. Fexinidazole is also under consideration as a new treatment for Chagas disease [78]. For benznidazole, *in vitro* activation also revealed reductive activation and ultimately disintegration of the compound to glyoxal [79]. Another study, however, failed to demonstrate the production of glyoxal in *T. cruzi*, although it revealed a plethora of metabolized products and adducts of these benznidazole breakdown-products [80]. This led to suggestions that, following its activation by *Tc*NTR1, benznidazole ultimately kills through the modification of numerous metabolite and protein targets in *T. cruzi* [80]. The metabolic fate of fexinidazole, following NTR1 mediated reduction, has not been determined, and the mechanism by which it kills trypanosomes is not known. However, a similar “cluster bomb” effect, with the activated fexinidazole product of the nitroreduction hitting multi-targets, seems likely. NTR1 has been proposed to act physiologically as an NADH dehydrogenase involved in reducing ubiquinone to ubiquinol [81], hence its essentiality even in bloodstream form *T. brucei* that requires ubiquinone-based electron transport in its mitochondrial alternative oxidase system [82]. In nifurtimox-treated *T. brucei*, loss of mitochondrial membrane potential and other mitochondrial morphology defects were noted [83]. Other nitroheterocycles, including fexinidazole, may also exert their effects on the mitochondrion, where the drug’s activation occurs. Knockout of a single copy of the *TbNTR1* gene (trypanosomes are diploid, hence have two copies of the gene, one on each copy of chromosome 7), led to diminished activity of the drug, albeit by just 1.6−1.9 fold, whilst double knockout was not possible [76]. Parasites resistant to fexinidazole could be selected *in vitro* under the conditions of increasing drug pressure (up to 20-fold resistance), and these parasites lost the 3’ flanking region of one allele of NTR, leading to a 50% reduction in the expression of the gene. The same study selected nifurtimox-resistant parasites, which lost one copy of the gene in developing 6-fold resistance. In both cases, resistance was significantly higher compared to the single NTR1 KO cells, indicating that other, as yet unknown, events beyond partial NTR1 loss, contribute to resistance. A question arises as to whether there is a risk that the current use of nifurtimox could lead to the selection of resistance to fexinidazole and vice versa. However, although diminished nitroreductase activity can yield cross resistance, it is not clear whether parasites, even with diminished nitroreductase, would be viable in the field, if the enzyme were critical in the insect transmitted forms of the parasites. Moreover, with very few cases of gambiense HAT currently reported (fewer than 1000 in 2018), the probability of selecting parasites resistant to either drug under field-settings is currently very low.

NTR1 is a typical enzyme of prokaryote origin. *Salmonella typhimurium*, the bacterium used in the Ames test, possesses nitroreductase genes as well, explaining the positive mutagenicity signal of fexinidazole and its metabolites in the Ames test. In contrast to *T. brucei*, however, nitroreductase activity is dispensable in *S. typhimurium*, and in a nitroreductase null mutant strain fexinidazole was no longer mutagenic [74]. Mammals do not have NTR1 orthologs, and thus, fexinidazole and fexinidazole sulfone were inactive in micronucleus tests with human or mouse cells [74].

## 4. Acoziborole: A Single Dose Oral Cure for Stage 2 HAT

As the enhanced effort to counter the surging HAT epidemic of the late twentieth century began, the prospect of an orally available drug that was able to cure stage 2, CNS-involved disease with a single administration seemed remote. Yet, as fexinidazole and pafuramidine were entering their first clinical trials, a novel, promising class of antimicrobial agents appeared on the scene: the benzoxaboroles [84]. These compounds are characterized by a core scaffold based around an oxaborole heterocycle fused to a benzene ring. Pursued by the small Californian company Anacor Pharmaceuticals Inc. (incorporated into Pfizer Inc. in 2016), the benzoxaboroles quickly attracted attention due to their numerous bioactivities. Activity against trypanosomes was first reported in 2010 [85], and DNDi selected this class for further work. The US-based pharmaceutical company Scynexis was contracted to initiate an intensive program of the structure-activity-relationship work to seek benzoxaboroles with good activity against trypanosomes. Crucially, these benzoxaboroles were also to demonstrate pharmacokinetic properties suitable to cure stage 2 HAT; i.e., being capable of reaching the CNS and retaining trypanocidal activity levels for long enough to obtain cure [86].

Cyrus Bacchi, the same investigator who had brought eflornithine forward for HAT in the 1980s [87], showed that a 6-carboxamido-based series cured the *T. brucei* strain GVR35 murine model of stage 2 disease. Structure-activity work eventually brought forward SCYX-7158, a gem-dimethyl 4-fluoro-2-trimethylfluoro benzamide derivative, which cured when given orally [88]. Its modest *in vitro* potency against *T. brucei* (IC_50_ around 0.6 μM) [88] was offset by good pharmacokinetic properties, giving a 100% cure in a mouse model of the stage 2 disease following a dosing of 25 mg/kg once a day for 7 days oral dosing [88]. Preclinical testing showed no overt toxicity in mice or dogs with a concentration of no observed adverse event limit (NOAEL) of 15 mg/kg. No binding to key proteins (CYP450), serine/cysteine peptidase or hERG channels emerged [89]. SCYX-7158 was also non-mutagenic in the Ames tests or standard mammalian cell genotoxicity assays [89]. Hence, phase I clinical safety trials were approved, and in 2012, SCYX-7158, or acoziborole, became the first new chemical entity resulting from DNDi’s program to enter clinical trials for HAT. 

A phase I study that included 128 healthy male subjects of sub-Saharan African origin was conducted in 2015 in France to assess safety, tolerability, pharmacokinetics and pharmacodynamics after single oral, ascending doses [90]. Some adverse effects were noted, including headaches, dizziness and GI tract reactions, as abdominal pain, nausea, vomiting, constipation and diarrhea. However, more serious issues were not identified. This trial led to the selection of a 960 mg dose, given as a single administration in three tablets (each with 320 mg of the active compound). Acoziborole’s long half-life (17 days), associated with high protein binding (97.8%) also necessitated safety monitoring in the volunteers for 210 days. Outputs from the phase I trials were good enough to enable progression to phase II/III trials in patients in Africa, which started late in 2016 [91]. The recruitment and dosing of patients is now complete, and follow-up is underway. Final results are eagerly awaited, but early indications (unpublished) suggest that the drug has demonstrated remarkable efficacy and safety profiles. 

### Acoziborole: Mode of Action and Resistance Risk

The mode of action and resistance mechanisms to benzoxaboroles are emerging. Acoziborole resistance was associated with multiple genetic changes in trypanosomes [92], although it was not possible to assign a specific gene to resistance in that study. However, among the changes in resistant lines was an amplification in the gene copy number of the RNA cleavage and polyadenylation specificity factor subunit 3 (CPSF3) [93]. Subsequently, CPSF3 was identified as a target for benzoxaboroles in apicomplexan parasites *Plasmodium* [93] and *Toxoplasma* [94]. Based on this observation, and the fact that metabolomics experiments had revealed a profound change in the methionine metabolism [95] that might have been related to RNA processing defects, particularly given the multi-methylation of the spliced leader sequence used in trans-splicing in trypanosomes, the effect of over-expression of CPSF3 on sensitivity to the related benzoxaborole, AN7973, was tested [96]. 

Over-expression of the gene yielded a notable loss of activity [96]. Elegant experiments using a gene over-expression library, followed by a selection of clones over-expressing genes yielding reduced sensitivity to acoziborole itself, also identified CPSF3 [97], and over-expression confirmed loss of sensitivity, pointing to CPSF3 as a target, if not the exclusive target, of these compounds in trypanosomes. 

Another trypanocidal benzoxaborole (of the amino-methyl subclass) was shown to be subject to a two-step metabolic processing, involving a primary conversion using amine oxidase in host serum to an aldehyde that is further metabolized to a carboxylate via parasite aldehyde dehydrogenase [98]. Another valinate–amide derivative series shows considerable promise for animal African trypanosomiasis too [99]. 

## 5. Conclusions

The twentieth century ended with human African trypanosomiasis epidemics raging across Africa. A disease that had been brought under control in the 1960s was wreaking havoc. This led to a concerted effort across a range of international organizations, including the WHO and newly formed agencies including DNDi and the CPDD. The arrival of the Bill and Melinda Gates Foundation and a re-alignment of many international aid efforts provided key investment to allow the development of new routes to combat HAT. The development of new drugs was central to the process. CPDD produced an orally available cure for stage 1 HAT that ultimately failed. In the meantime, DNDi brought forward an old nitroheterocycle, fexinidazole, and through the combined forces of persistence and diligence worked their way around numerous perceptual hurdles related to the nitroheterocycle class, so as to bring to the clinic the first wholly oral treatment for stage 1 and 2 HAT. As those trials were proceeding, DNDi created a consortium to bring forward a class of compounds, the benzoxaboroles, from Anacor Pharmaceuticals Inc., through to Scynexis and an extended collaborative team. Encouraging clinical results for the leading candidate compound, acoziborole, now promise to deliver a drug that may cure stage 2 HAT with a single oral dose. The ability of non-profit Product Development Partnerships to bring new medicines to market is now proven. The turnaround in the epidemiology of HAT in the twenty first century is a truly remarkable story and a classic model of how to create medical success.

## Figures and Tables

**Figure 1 tropicalmed-05-00029-f001:**
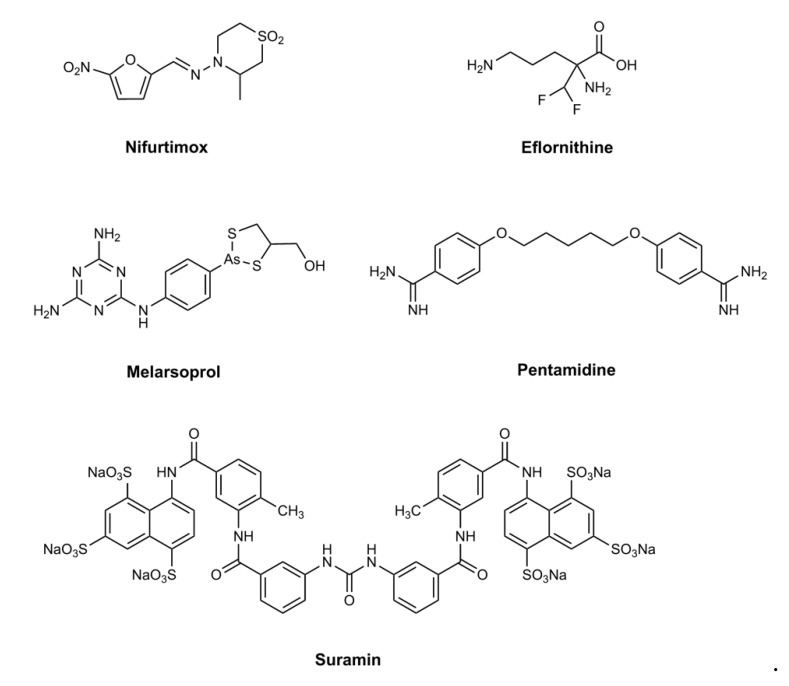
Different drugs have been used to treat HAT depending on the trypanosome subspecies causing the disease, and whether progression is at stage 1 or 2. For the past decade nifurtimox and eflornithine combination therapy has been the treatment of choice for stage 2 *T. b. gambiense* disease, and pentamidine for stage 1. For rhodesiense HAT, stage 2 is treated with melarsoprol and stage 1 with suramin.

**Figure 2 tropicalmed-05-00029-f002:**
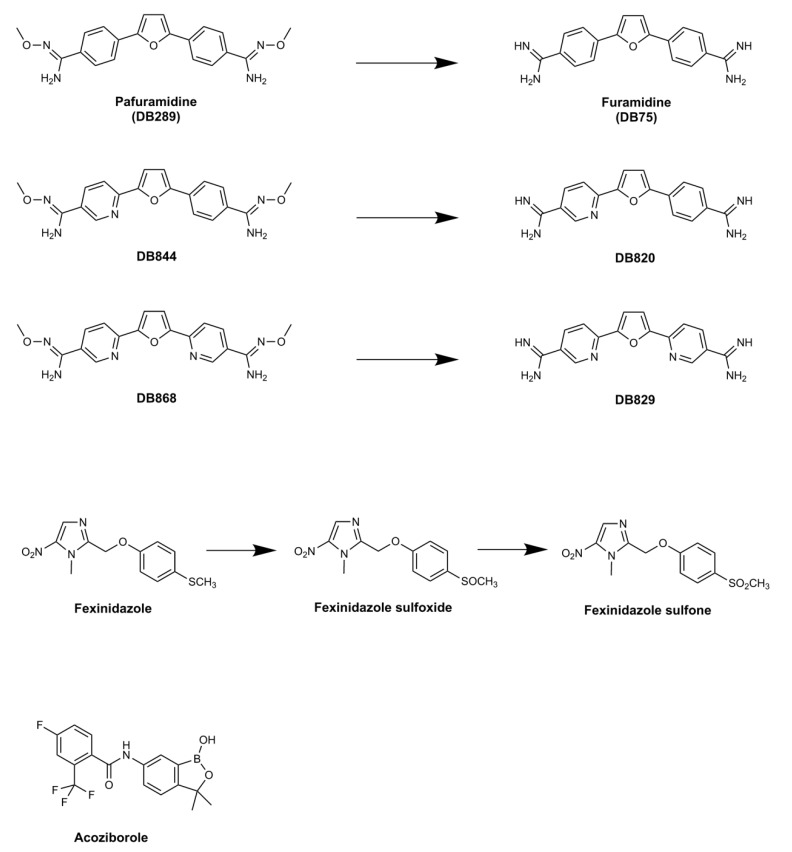
Clinical trials with pafuramidine (DB289), which is the prodrug of furamidine (DB75), failed due to the appearance of renal toxicity during extended phase I safety profiling. The aza-derivatives including DB868—a prodrug of DB829, and DB844—a prodrug of DB820, also showed activity against stage 2 disease, but development was halted following identification of the toxicity associated with pafuramidine. Fexinidazole has been approved for use by the European Medicines Agency in 2018. The compound is converted to sulfoxide then sulfone derivatives after administration. Acoziborole is a benzoxaborole in clinical trials, where the efficacy of a single dosing of the drug as an oral medication against stage 2 disease is being assessed.

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
