# Peer review of "New Drugs for Human African Trypanosomiasis: A Twenty First Century Success Story"

_tropicalmed, 2020, doi:10.3390/tropicalmed5010029_

Round 1
Reviewer 1 Report
This is a well-written short review about the recent successes in drug development for Human African Trypanosomiasis. The review, with as authors several prominent actors in this area from different institutions, is informative and up to date; it is a highly suitable contribution to MDPI’s ‘Tropical Medicine and Infectious Disease’ special issue devoted to HAT.
I have only a few very minor comments/suggestions:
Title 1: It would be nicer to move the a at the end of line 1 to the start of line 2. Lines 3-4: Affiliation e (mentioned in line 10) is missing (should it be with Srinivasa Rao instead of d?) Line 51: Since Médecines Sans Frontières is a French name of the organization, please add the accents. Line 131: There seems no need to use upper case W for world. Line 153: Should this not be ‘7% increase in release of liver enzymes’? Similarly for line 164. Lines 200 and 203: the 0 in F1F0 should be an o, not zero – the o stands for oligomycin (the ATPase subunit with sensitivity oligomycin, as originally named so by Racker and Kagawa). Line 214: the comma present after ‘being’ should be placed in front of it. Line 268: italicize in vitro Line 272: Reference 74 about the safety of Fexinidazole dates from 2012, well before clinical trials had been completed. I wonder if it would not be appropriate to mention here briefly the result of safety assessments from the trials and add also a more recent reference that lists the advantages and limitations of the drug. For example: Pelfrene E et al. (2019) The European Medicines Agency’s scientific opinion on oral fexinidazole for human African trypanosomiasis. PLoS Negl Trop Dis 13 (6): e0007381. Alternatively, if this will be discussed in a different paper of this special issue, refer to it. Line 338: Maybe for non-expert readers: explain NOAEL Line 378: delete 1x centuryAuthor Response
REVIEWER 1:
This is a well-written short review about the recent successes in drug development for Human African Trypanosomiasis. The review, with as authors several prominent actors in this area from different institutions, is informative and up to date; it is a highly suitable contribution to MDPI’s ‘Tropical Medicine and Infectious Disease’ special issue devoted to HAT.
I have only a few very minor comments/suggestions:
Title 1: It would be nicer to move the a at the end of line 1 to the start of line 2. DONE
Lines 3-4: Affiliation e (mentioned in line 10) is missing (should it be with Srinivasa Rao instead of d?) YES IT IS “e”. THANKS, WE HAVE CHANGED IT
Line 51: Since Médecines Sans Frontières is a French name of the organization, please add the accents.
CHANGED AS REQUESTED
Line 131: There seems no need to use upper case W for world.
DROPPED IN LINE WITH REQUEST FROM ANOTHER REFEREE
Line 153: Should this not be ‘7% increase in release of liver enzymes’?
NO, IT IS THE 7% RISE IN INDIVIDUALS SOWING INCREASED LIVER ENZYMES THAT WE REFER TO AND BELIEVE THIS IS AS IT IS STATED
Similarly for line 164. Lines 200 and 203: the 0 in F1F0 should be an o, not zero – the o stands for oligomycin (the ATPase subunit with sensitivity oligomycin, as originally named so by Racker and Kagawa).
THANK YOU. CHANGED AS REQUESTED
Line 214: the comma present after ‘being’ should be placed in front of it.
CHANGED
Line 268: italicize in vitro
CHANGED
Line 272: Reference 74 about the safety of Fexinidazole dates from 2012, well before clinical trials had been completed. I wonder if it would not be appropriate to mention here briefly the result of safety assessments from the trials and add also a more recent reference that lists the advantages and limitations of the drug. For example: Pelfrene E et al. (2019) The European Medicines Agency’s scientific opinion on oral fexinidazole for human African trypanosomiasis. PLoS Negl Trop Dis 13 (6): e0007381. Alternatively, if this will be discussed in a different paper of this special issue, refer to it. Line 338: Maybe for non-expert readers: explain NOAEL Line 378: delete 1x century
WE HAVE ADDED REFERENCE 67 (THE FEXINIDAZOLE ARTICLE INCLUDED IN THE SPECIAL ISSUE
Reviewer 2 Report
In the review article entitled “New drugs for human African trypanosomiasis: a twenty first century success story” Emily A. Dickie and co-workers have compiled recent advancements made in the development of new drugs to cure African trypanosomiasis. The authors have written this review very systematically and covered different aspects of each drug. The only thing which I felt missing is that the authors have not given much emphasis on the mechanistic aspects of these drugs and have not discussed the targets proteins. It would be better if authors can add some more information on the mechanistic aspect of these drugs.
Author Response
The revision by E. A. Dickie et al has been written very well and I find it very useful to have an updated view of the drugs used against HAT in the 21st century:
Please find below a list of minor details to improve the manuscript:
- a space is missing in the title, says “New drugs for human African trypanosomiasis:a twenty first century success story” bu t should say “New drugs for human African trypanosomiasis: a twenty first century success story”.
CHANGED AS REQUESTED
- in page 2 says “Medecins Sans Frontieres (MSF)” but should say “Médecins Sans Frontières (MSF)”.
CORRECTED
- in page 2 row 48 a space is missing: says “infected [9].In response,” but should say “infected [9]. In response,”.
CORRECTED
- also in page 2 row 76 a space is missing: says “(NITD) [22].Small” but should say “(NITD) [22]. Small”.
CORRECTED
- captions of figure 1 and 2 are repeated: one at the front and another one in the back of each figure. Only one caption should be indicated.
WE COULDN’T SEE THIS. PERHAPS THE EXITORAL OFFICE ALREDY CORRECTED?
- in caption of Figure 2, authors might want to mention something about DB844 and DB820 drugs.
THIS HAS BEEN ADDED
-if pentamidine is a di-cation as said in page 4, then the chemical structure of pentamidine depicted in Figure 1 better to be indicated as a di-cation.
THE GIVEN STRUCTURE IS THE CONVENTIONAL ONE (WITHOUT POSITIVE CHARGES INDICATED)
- authors might want to indicate teh meaning of “b.i.d.” when says “(100 mg b.i.d.)” in page 5 row 138.
CORRECTED
- sometimes is used “BID” term (page 148 row 148) and sometimes “b.i.d.”, only one should be used along the text.
WE CHANGED TO b.i.d. THROUGHOUT
- a space is missing in page 9 row 315: says “[74].Mammals” but should say “[74]. Mammals”.
CORRECTED
- in row 353, letters are bigger.
CORRECTED
- in page 10 row 378 says “The twentieth centurycentury” but should say “The twentieth century”.
CORRECTED
Reviewer 3 Report
The revision by E. A. Dickie et al has been written very well and I find it very useful to have an updated view of the drugs used against HAT in the 21st century:
Please find below a list of minor details to improve the manuscript:
- a space is missing in the title, says “New drugs for human African trypanosomiasis:a twenty first century success story” bu t should say “New drugs for human African trypanosomiasis: a twenty first century success story”.
- in page 2 says “Medecins Sans Frontieres (MSF)” but should say “Médecins Sans Frontières (MSF)”.
- in page 2 row 48 a space is missing: says “infected [9].In response,” but should say “infected [9]. In response,”.
- also in page 2 row 76 a space is missing: says “(NITD) [22].Small” but should say “(NITD) [22]. Small”.
- captions of figure 1 and 2 are repeated: one at the front and another one in the back of each figure. Only one caption should be indicated.
- in caption of Figure 2, authors might want to mention something about DB844 and DB820 drugs.
-if pentamidine is a di-cation as said in page 4, then the chemical structure of pentamidine depicted in Figure 1 better to be indicated as a di-cation.
- authors might want to indicate teh meaning of “b.i.d.” when says “(100 mg b.i.d.)” in page 5 row 138.
- sometimes is used “BID” term (page 148 row 148) and sometimes “b.i.d.”, only one should be used along the text.
- a space is missing in page 9 row 315: says “[74].Mammals” but should say “[74]. Mammals”.
- in row 353, letters are bigger.
- in page 10 row 378 says “The twentieth centurycentury” but should say “The twentieth century”.
Author Response
In the review article entitled “New drugs for human African trypanosomiasis: a twenty first century success story” Emily A. Dickie and co-workers have compiled recent advancements made in the development of new drugs to cure African trypanosomiasis. The authors have written this review very systematically and covered different aspects of each drug. The only thing which I felt missing is that the authors have not given much emphasis on the mechanistic aspects of these drugs and have not discussed the targets proteins. It would be better if authors can add some more information on the mechanistic aspect of these drugs.
WE HAVE INCLUDED EXTENSIVE SECTIONS ON WHAT IS KNOWN ON MECHANISM OF EACH OF THE DRUGS DISCUSSED ALREADY (A DEDICAETD SECTION IS ASSOCIAETD WITH EACH)
Reviewer 4 Report
The review by Dickie et al is well-written overall, and provides an informative overview of the development of pafuramidine, fexinidazole, and acoziborole, and what has been learned from laboratory studies about MoA and drug resistance.
The stated purpose of the review (24-26) is to “describe the remarkable successes in combatting HAT through the 21st century”, but it reads as history of drug development efforts of the 3 aforementioned compounds, with the development of pafuramidine, receiving by far the most detailed treatment. Section 2.1, providing a very detailed dissection of the pafuramidine clinical trial results, is misplaced here, and should be greatly reduced. Speculation about why the delayed renal insufficiency was observed in the 2nd phase I trial, the promise of other diamidine analogues in animal models, etc. is not in keeping with the stated purpose of the review.
The MS could be improved by addressing a few issues, detailed below.
The introduction should provide a concise introduction to HAT and the causative agents, clearly delineating key differences between b. rhodesiense vs. T. b. gambiense, and relevance to elimination prospects and strategies. Currently, key information is lacking, dropped parenthetically, included in figure legends, etc., which a non-specialist would need for context.
1 (“current drugs”) vs. Fig. 2 (“new drugs”) chould be adapted—perhaps “historical” vs. “21st century”? Fexinidazole (which should become current now, in 2020) could be differentiated from the clinical candidates (halted and current), and the excess of diamidine analogues could be removed.
The name/title/credentials dropping in the pre-clinical/clinical development sections could be reduced, or better, eliminated. It contrasts sharply with the MoA and resistance sections, in which work is simply described and cited without naming the scientists who led the projects.
Introducing the organisations involved in the research once, then abbreviating, should be sufficient, e.g. CPDD is defined and abbreviated on pages 2, 5, 10.
The impact and lessons learned from pafuramidine clinical trial design/implementation can rightfully be highlighted. However, the authors should take care in their language here. Hyberbole as in lines 131-132: “…in a difficult cohort of patients in some of the World’s least hospitable places to work” while surely unintentional, can be read as an indictment of the very people that these medicines are being developed for. A concise description of the actual challenges overcome to conduct these trials should be included instead.
Typesetting should be checked throughout: e.g. lines 1, 377 are missing spaces. “centurycentury” typo in Line 378.
Author Response
The stated purpose of the review (24-26) is to “describe the remarkable successes in combatting HAT through the 21st century”, but it reads as history of drug development efforts of the 3 aforementioned compounds, with the development of pafuramidine, receiving by far the most detailed treatment. Section 2.1, providing a very detailed dissection of the pafuramidine clinical trial results, is misplaced here, and should be greatly reduced. Speculation about why the delayed renal insufficiency was observed in the 2nd phase I trial, the promise of other diamidine analogues in animal models, etc. is not in keeping with the stated purpose of the review.
WE APPRECIATE THE REVIEWER’S CONCERN, HOWEVER WE PREFER TO RETAIN THIS SECTION AS WE BELIEVE THE PAFURAMIDINE STORY PLAYED AN IMPORTANT ROLE IN PAVING THE WAY FOR OTHER DRUG DEVELOPMENT PROGRAMMES FOR HAT – AND UNDERSTANDING THE PROBLEMS BEHIND THAT DRUG’S FAILURE WAS A KEY PART OF THE PROCESS.
The MS could be improved by addressing a few issues, detailed below.
The introduction should provide a concise introduction to HAT and the causative agents, clearly delineating key differences between b. rhodesiense vs. T. b. gambiense, and relevance to elimination prospects and strategies. Currently, key information is lacking, dropped parenthetically, included in figure legends, etc., which a non-specialist would need for context.
WE HAVE ADDED SUCH A SECTION
1 (“current drugs”) vs. Fig. 2 (“new drugs”) chould be adapted—perhaps “historical” vs. “21stcentury”? Fexinidazole (which should become current now, in 2020) could be differentiated from the clinical candidates (halted and current), and the excess of diamidine analogues could be removed.
FIGURE TITLES HAVE BEEN AMENDED IN LINE WITH THE REQUEST
The name/title/credentials dropping in the pre-clinical/clinical development sections could be reduced, or better, eliminated. It contrasts sharply with the MoA and resistance sections, in which work is simply described and cited without naming the scientists who led the projects.
WE HAVE REMOVED THOSE NAMES
Introducing the organisations involved in the research once, then abbreviating, should be sufficient, e.g. CPDD is defined and abbreviated on pages 2, 5, 10.
CORRECTED THROUGHOUT
The impact and lessons learned from pafuramidine clinical trial design/implementation can rightfully be highlighted. However, the authors should take care in their language here. Hyberbole as in lines 131-132: “…in a difficult cohort of patients in some of the World’s least hospitable places to work” while surely unintentional, can be read as an indictment of the very people that these medicines are being developed for. A concise description of the actual challenges overcome to conduct these trials should be included instead.
WE REMOVED THAT PARTICULAR HYPERBOLE WHICH INDEED WAS NOT AIMING TO OFFEND PATIENTS!
Typesetting should be checked throughout: e.g. lines 1, 377 are missing spaces. “centurycentury” typo in Line 378.
CORRECTED, ALONG WITH OTHER INSTANCES OUTLINED ABOVED.